# Metabolic reconstitution of germ-free mice by a gnotobiotic microbiota varies over the circadian cycle

Daniel Hoces[1], Jiayi Lan[2], Wenfei Sun[3¤], Tobias Geiser[1], Melanie L. Stäubli[4], Elisa Cappio Barazzone[1], Markus Arnoldini[1], Tenagne D. Challa[3], Manuel Klug[3], Alexandra Kellenberger[3], Sven Nowok[5], Erica Faccin[1], Andrew J. Macpherson[6], Bärbel Stecher[7,8], Shinichi Sunagawa[4], Renato Zenobi[2], Wolf-Dietrich Hardt[4], Christian Wolfrum[3], Emma Slack[1,9]*

1 Laboratory for Mucosal Immunology, Institute of Food, Nutrition and Health, Department of Health Sciences and Technology, ETH Zürich, Zürich, Switzerland, 2 Laboratory of Organic Chemistry, Department of Chemistry and Applied Biosciences, ETH Zürich, Zürich, Switzerland, 3 Laboratory of Translational Nutrition Biology, Institute of Food, Nutrition and Health, Department of Health Sciences and Technology, ETH Zürich, Schwerzenbach, Switzerland, 4 Institute of Microbiology, Department of Biology, ETH Zürich, Zürich, Switzerland, 5 ETH Phenomics Center, Department of Biology, ETH Zürich, Zürich, Switzerland, 6 Department of Visceral Surgery and Medicine, Bern University Hospital, University of Bern, Bern, Switzerland, 7 Max-von-Pettenkofer Institute, LMU Munich, Munich, Germany, 8 German Center for Infection Research (DZIF), partner site LMU Munich, Munich, Germany, 9 Botnar Research Centre for Child Health, Basel, Switzerland

¤ Current address: Stanford University, Department of Bioengineering, Stanford, California, United States of America
* emma.slack@hest.ethz.ch

**Data Availability Statement:** Relevant source data for Figs 1C–1G, 2A–2I, 3A–3F, 4A–4F, and S1A–S1D, S2A–S2K, S3A–S3E, S4, S5A–S5C, S6, S7,

## Abstract

The capacity of the intestinal microbiota to degrade otherwise indigestible diet components is known to greatly improve the recovery of energy from food. This has led to the hypothesis that increased digestive efficiency may underlie the contribution of the microbiota to obesity. OligoMM12-colonized gnotobiotic mice have a consistently higher fat mass than germ-free (GF) or fully colonized counterparts. We therefore investigated their food intake, digestion efficiency, energy expenditure, and respiratory quotient using a novel isolator-housed metabolic cage system, which allows long-term measurements without contamination risk. This demonstrated that microbiota-released calories are perfectly balanced by decreased food intake in fully colonized versus gnotobiotic OligoMM12 and GF mice fed a standard chow diet, i.e., microbiota-released calories can in fact be well integrated into appetite control. We also observed no significant difference in energy expenditure after normalization by lean mass between the different microbiota groups, suggesting that cumulative small differences in energy balance, or altered energy storage, must underlie fat accumulation in OligoMM12 mice. Consistent with altered energy storage, major differences were observed in the type of respiratory substrates used in metabolism over the circadian cycle: In GF mice, the respiratory exchange ratio (RER) was consistently lower than that of fully colonized mice at all times of day, indicative of more reliance on fat and less on glucose metabolism. Intriguingly, the RER of OligoMM12-colonized gnotobiotic mice phenocopied fully colonized mice during the dark (active/eating) phase but phenocopied GF mice during the light (fasting/resting)

S8 is available in S1 Data. Raw sequencing data used for S6 Fig is publicly available on the European Nucleotide Archive (ENA) under the Project ID PRJEB53981. Raw data and code used for generating all figures in this publication are made available in a curated data archive at ETH Zurich (https://www.research-collection.ethz.ch/handle/20.500.11850/521803) under the DOI 10.3929/ethz-b-000521803.

**Funding:** This work was funded by the Novartis FreeNovation (https://www.novartis.ch/de/novartis-in-der-schweiz/medizin-neu-denken/forschungsfoerderung/freenovation) (E.S., W-D.H., C.W.); NCCR Microbiomes, a research consortium financed by the Swiss National Science Foundation (https://nccr-microbiomes.ch) (E.S., W-D.H, S.S.); Swiss National Science Foundation (40B2-0_180953, 310030_185128; https://www.snf.ch/en) (E.S.), European Research Council Consolidator Grant (NUMBER 865730-SNUGly; https://erc.europa.eu/funding/consolidator-grants) (E.S.), Gebert Rüf Microbials (GR073_17; https://www.grstiftung.ch/en/area-activity/closed-areas/microbials.htm) (E.S.); Botnar Research Centre for Child Health Multi-Invesitigator Project 2020 (BRCCH_MIP: Microbiota Engineering for Child Health; https://brc.ch) (E.S.), ETH Zürich Foundation and Evi Diethelm-Winteler-Stiftung (Zurich Exhalomics; https://www.exhalomics.ch) (R.Z.). The funders had no role in study design, data collection and analysis, decision to publish, or preparation of the manuscript.

**Competing interests:** The authors have declared that no competing interests exist.

**Abbreviations:** GF, germ-free; iBAT, interscapular brown adipose tissue; iWAT, inguinal white adipose tissue; RER, respiratory exchange ratio; SCFA, short-chain fatty acid; SPF, specific-opportunistic-pathogen-free; UPLC/MS, ultraperformance liquid chromatography coupled with mass spectrometry; vWAT, visceral white adipose tissue; ZT, Zeitgeber time.

phase. Further, OligoMM12-colonized mice showed a GF-like drop in liver glycogen storage during the light phase and both liver and plasma metabolomes of OligoMM12 mice clustered closely with GF mice. This implies the existence of microbiota functions that are required to maintain normal host metabolism during the resting/fasting phase of circadian cycle and which are absent in the OligoMM12 consortium.

## Introduction

The gut microbiota is currently considered a key regulator of host energy metabolism [1]. In the absence of a microbiota, mice accumulated less fat [2] and were protected from obesity induced by certain types of high-fat diets [3–5]. Several mechanisms have been proposed to explain this phenomenon and its relationship to metabolic imbalances [6]. These include endocrine regulation of food intake [7,8], additional energy liberated by the microbiota from dietary fibers [9], alterations in bile acid profiles [10,11], inflammatory responses induced by some members of the microbiota [12], and induction of thermogenesis in adipose tissue [13–15]. However, given the complexity of a complete microbiota and its interactions with the host, validating any of these theories and identifying causal relationships remains a major experimental challenge [16,17].

Gnotobiotic mice, colonized with a simplified microbiota made up of defined species, have become a major tool to identify potential mechanisms of interaction between the microbiota and host [18–20]. Such approaches can generate a mechanistic understanding of how external factors (i.e., diet, infection) act on the different microbiota members individually and at a community level [21,22]. A widely used example, the OligoMM12, is a gnotobiotic consortium of 12 cultivable mouse-derived strains representing the major 5 bacterial phyla in the murine gut [23]. It is reproducible between facilities [24] and extensive data now exist on the metabolism of individual species and their metabolic interactions with each other [25–28]. Understanding how and to what extent, this gnotobiotic microbiota reconstitutes the metabolic phenotype of conventional mice is therefore of broad relevance for microbiota research.

Circadian variations in microbiota function adds an extra layer of complexity to metabolic interactions between the host and the microbiota. Circadian feeding is a major driver of microbiota composition [29,30]. The luminal concentration of fermentation products such as short-chain fatty acids (SCFAs) shows a dramatic circadian oscillation linked both to food intake and to intestinal motility [31]. Microbiota-derived molecules are known to influence host nutrient absorption [32] and host metabolic gene expression [33,34]. However, much of our current knowledge is derived from indirect calorimetry measurements made over a time period shorter than 24 h [2,3,35,36]. Measurements of the same host–microbiota system, if taken at different time points in the circadian cycle of metabolism, could therefore be wrongly interpreted as qualitative shifts in microbiota function. Consequently, to understand the influence of the microbiota on host energy metabolism, it is key to quantify variation over the full circadian cycle.

A challenging aspect of addressing the influence of the OligoMM12 microbiota on host metabolism is that long-term experiments require hygiene barrier conditions similar to those required to work with germ-free (GF) mice. In particular, standard metabolic cage systems do not permit maintenance of an axenic environment, and moving mice between the open cages typically used in isolator systems where such animals are normally bred, to IVC cage-like systems used for most metabolic cages, can be associated with stress and behavioral abnormalities

[37]. We have therefore built an isolator-housed metabolic cage system. Based on the TSE PhenoMaster system, we can monitor levels of $O_2$, $CO_2$, and hydrogen every 24 min for up to 8 cages, across 2 separate isolators in parallel, while maintaining a strict hygienic barrier. With this custom-built system, longitudinal monitoring of metabolism can be carried out over periods of several weeks in GF and gnotobiotic mice.

In this study, we applied isolator-housed indirect calorimetry to understand how well gnotobiotic microbiota replicate the influence of a complex microbiota on host metabolism. We compared the metabolic profile of GF, gnotobiotic (OligoMM12), and conventionally raised mice (specific-opportunistic-pathogen-free (SPF)) fed ad libitum with standard chow. This revealed the potential for gnotobiotic mouse systems to identify microbiota species and functions essential to support normal host metabolism.

## Results

To compare to published literature on GF and colonized mouse metabolism, we compared male, adult age-matched (12 to 14 wk old) GF, gnotobiotic (OligoMM12), and conventionally raised (SPF) mice, all bred and raised in flexible-film isolators and with a C57BL/6J genetic background. Indirect calorimetry measurements were carried out in flexible-film surgical isolators accommodating a TSE PhenoMaster system (Fig 1A and 1B). Mice were adapted for between 24 and 36 h to the single-housing condition inside isolator-based metabolic chambers before data collection. Variations on $O_2$, $CO_2$, and hydrogen, along with food and water consumption, were recorded every 24 min on each metabolic cage. We could confirm that GF mice maintain their GF status over at least 10 d of accommodation in these cages, via culture-dependent and culture-independent techniques (S2A–S2D Fig).

### Body composition in GF, OligoMM12, and SPF mice

After data recording for indirect calorimetry, mice were fasted for 4 to 5 h and killed (approximately at Zeitgeber time (ZT) 6 ± 1 h), and body mass and body composition were measured. As cecal mass (cecal tissue plus its content) is massively affected by the colonization status [35], we first assessed the cecal mass in GF, OligoMM12, and SPF and its impact on body mass. We found that cecal mass was inversely correlated to the microbiota complexity, starting at approximately 0.5 g in SPF mice, increasing to around 1.5 g in OligoMM12 mice and reaching 3 g on average in GF mice (Fig 1C). Note that this represents around 10% of total body mass in GF mice (S2A Fig), which translates into a trend to increased total body mass in GF mice (Fig 1D). This trend was completely reverted after removal of the cecum from total mass (Fig 1E).

Measurements of body composition in mice are often performed using EchoMRI, which yields data on lean, fat, and water mass. We observed a nonsignificant increasing trend in lean body mass from GF to SPF mice (Fig 1F). GF mice had a significantly lower percentage of lean body mass than colonized mice (S2B Fig). As cecal content water retention can contribute up to 10% of the total body weight of a GF mouse (Fig 1C), we hypothesized that this would be the major contributor to a lower percentage lean mass. However, EchoMRI readouts of fat mass seemed inconsistent with this assumption. We therefore compared EchoMRI readouts of "lean" and "fat" body mass before and after removal of the cecum. We found a strong correlation between the total lean mass measured by EchoMRI with and without the cecum (S2C Fig), i.e., cecum removal consistently reduced the lean mass readout by 5% to 10% (S2D and S2E Fig). Therefore, cecum removal has a relatively consistent effect on lean mass across groups. For ease of comparison to published work, we decided to use lean mass obtained by EchoMRI before dissection for definitive energy expenditure calculations.

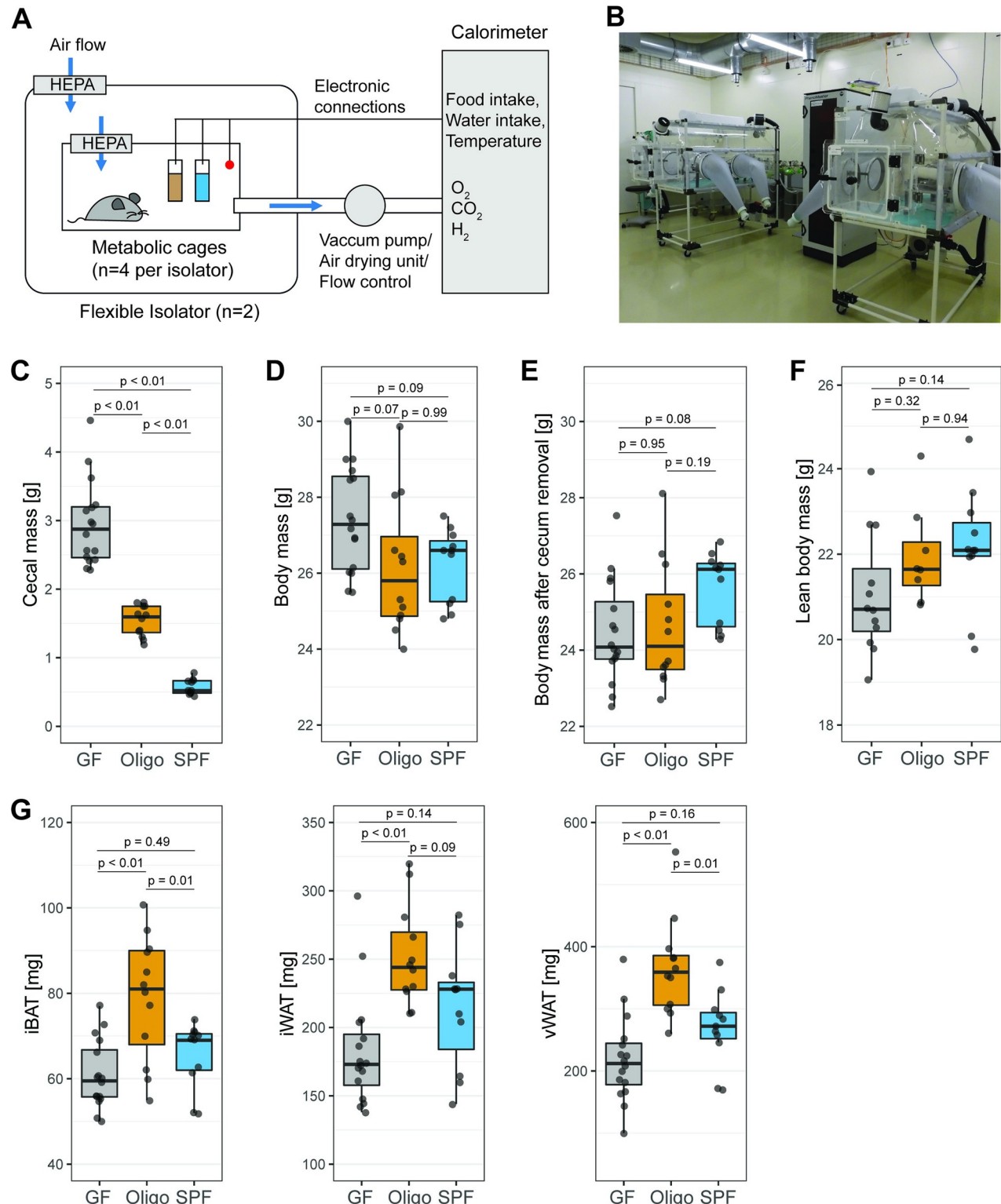

**Fig 1. OligoMM12 mice have increased fat mass compared to GF mice and SPF C57B6/J mice.** (A) Schematic representation of isolator-based indirect calorimetry system, with a TSE PhenoMaster calorimeter connected to 2 flexible surgical isolators with 4 metabolic cages each. (B) Pictures of isolator-based indirect calorimetry system inside the facility. (C) Cecal mass (tissue including luminal content). (D) Total body mass at the end of the experiment and before cecum removal. (E) Total body mass after cecum removal. (F) Lean body mass acquired by EchoMRI before cecum removal (N of mice per group with EchoMRI and indirect calorimetry measurements: GF = 12, OligoMM12 = 8, SPF = 11). (G) Fat mass from iBAT, iWAT, and

vWAT. Number of mice per group in all figures unless otherwise specified: GF = 16, OligoMM12 = 12, SPF = 11. $p$-values obtained by Tukey's honest significance test. Data underlying this figure are supplied in S1 Data. GF, germ-free; iBAT, interscapular brown adipose tissue; iWAT, inguinal white adipose tissue; SPF, specific-opportunistic-pathogen-free; vWAT, visceral white adipose tissue.

In contrast, EchoMRI fat mass measurements pre- and post-cecum dissection were poorly correlated in GF mice (S2F–S2H Fig) attributable to a highly variable percentage scoring of cecal content as either fat or water. As total fat mass is in the order of 2 to 4 g and the cecum of a GF mouse can easily have a mass of 3 g (Fig 1C), it is clear that aberrantly scoring 50% of the cecum as "fat" will have a massive impact on the EchoMRI-measured "fat mass". Correspondingly, in GF mice, cecum removal resulted in a decrease in EchoMRI fat mass readout of between 5% and 48% (S2I and S2J Fig). Worryingly, we also observed a shift towards higher fat mass readings in SPF mice after cecum removal (S2I and S2J Fig), which occurred over and above the known phenomena of inaccuracies in fat mass estimation when comparing live and dead animals [38,39] (S2K Fig). In summary, these results further highlighting the need for caution in interpreting EchoMRI readouts for fat mass in mice with major differences in intestinal composition. Therefore, we proceeded to directly weigh the fat depots accessible to dissection (interscapular brown adipose tissue (iBAT); and inguinal and visceral white adipose tissue (iWAT and vWAT)). There was no significant difference between GF and SPF mice in size of the explored fat depots; however, OligoMM12 mice accumulated more fat in all explored depots than GF mice, including more iBAT and vWAT, compared to SPF mice (Fig 1G).

## Energy metabolism and energy balance in GF, OligoMM12, and SPF mice

Body composition is determined by the quantity of calories absorbed from food and whether these calories are directly expended or are stored. Energy expenditure was estimated using $VO_2$ and $VCO_2$ readouts [40] and normalized as described before [41–43] using EchoMRI lean body mass (Fig 1F) and dissected fat mass (Fig 1G).

As described before, energy expenditure showed a linear relation with lean body mass (Fig 2A) and varied over the circadian cycle (S3A Fig). Although raw energy expenditure appears higher in SPF mice (S3B Fig), this difference disappears on normalization using a regression model that included lean body mass and total dissected fat mass as predictive variables (Fig 2B). This lack of difference was also observed when light and dark phase were analyzed separately (Fig 2B). "Classical" normalization procedures (dividing by mass) also showed no difference between groups when "total body mass after cecum dissection" (S3C Fig) or lean body mass (S3D Fig) was used for normalization of energy expenditure. Unsurprisingly, we did calculate a significant difference during the dark phase in energy expenditure between GF and SPF mice if "total body mass" was used for normalization (S3E Fig), which is an artefact attributable to the inclusion of around 10% extra body mass in the GF mice, contributed by inert cecal water. Therefore, at least when comparing to the SPF microbiota used in this study, absence of a microbiota does not result in altered daily energy expenditure in metabolically active tissues.

We next investigated calorie absorption from food by comparing the daily energy ingestion from food and calorie excretion in feces of GF, OligoMM12, and SPF mice. The difference between these 2 values estimates the absorbed calories. As reported previously [44], GF animals ingested on average between 10% and 20% more standard chow compared to OligoMM12 and SPF mice (Fig 2C). Correspondingly, GF animals also excreted a much larger dry mass of feces, while OligoMM12 mice produced an intermediate fecal mass and SPF mice excreted the least (Fig 2D).

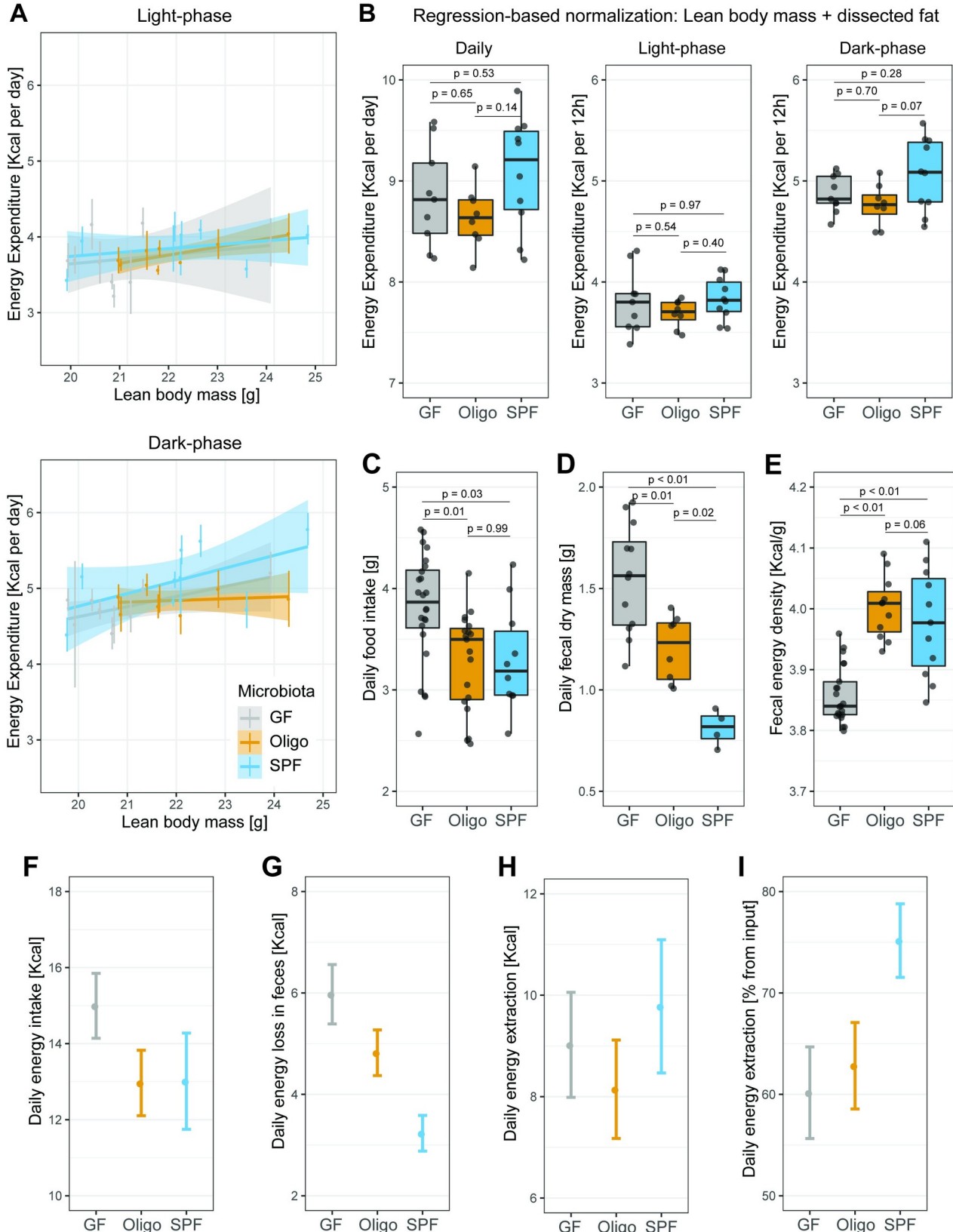

**Fig 2. Energy metabolism in GF, OligoMM12, and SPF C57B6/J mice.** (A) Linear regression of energy expenditure and lean body mass based on EchoMRI during light and dark phase. Each colored vertical line represents energy expenditure measurements during the experiment for 1 mouse. (B) Energy expenditure during 24-h period or during the 12-h light or dark phase. Values represent area-under-curve normalized by regression-based analysis using lean body mass obtained by EchoMRI and dissected fat mass. (C) Average daily food intake per mouse. Mice represented in this figure include those that underwent long-term indirect calorimetry (Fig 3) and mice that only contribute to daily fecal pellet quantification/bomb calorimetry. (N of mice per group: GF = 24, OligoMM12 = 19, SPF = 10) (D) Dry fecal output per mouse collected during a 24-h period. (N of mice per group: GF = 12, OligoMM12 = 8, SPF = 4) (E) Energy content of dry fecal output by bomb calorimetry. (N of mice per group: GF = 21, OligoMM12 = 11, SPF = 11). (F-I) Estimation energy metabolism parameters. Number represented estimate mean value ± 1.96*combined standard uncertainty from measurements used for calculations. (F) Estimated daily energy input (food intake* 3.94 kcal/g). (G) Estimated daily energy excretion (daily fecal dry mass*fecal energy content). (H) Estimated daily energy extraction (daily energy input–daily energy excretion). (I) Estimated energy extraction from food as percentage of energy input ((daily energy input – daily energy excretion)/daily energy input*100). Note that calculations in F, G, and H are per mouse and are not normalized to body mass. Number of mice per group in all figures unless otherwise specified: GF = 9, OligoMM12 = 8, SPF = 10. *p*-values obtained by Tukey's honest significance test. Data underlying this figure are supplied in S1 Data. GF, germ-free; SPF, specific-opportunistic-pathogen-free.

Remarkably, energy density of dry feces was lower in GF mice (3.7 kcal/g) compared to colonized mice (OligoMM12 and SPF, 4.0 kcal/g), with the latter showing no difference among them (Fig 2E). This gap between GF and mice with microbiota can likely be explained by the fact that although fecal bacteria improve energy release from food, a considerable fraction of that energy remains stored in the bacteria present in the feces. We measured bacterial density in the cecum content of OligoMM12 and SPF by bacterial flow cytometry (S4 Fig), which gave us a good estimation of bacterial density [45]. Using the average bacterial density per type of mice (OligoMM12 = $1.1 \times 10^{11}$ bacteria cells/g and SPF = $1.6 \times 10^{11}$ bacterial cells/g) and assuming certain parameters (dry mass of a bacterium = $2.26 \times 10^{-13}$ g/bacteria cell [46], and energy stored in bacteria = 4.58 kcal/g of dry bacteria mass [47]); we estimated that the fecal microbiota of colonized mice can contribute between 0.11 kcal/g of dry fecal mass in OligoMM12 to 0.17 kcal/g of dry fecal mass in SPF—which is in the range of energy density difference between fecal energy density in colonized and GF mice.

We then used these values for food intake, fecal dry mass output, and fecal energy density to estimate energy absorbed from the feces. We found that the higher food consumption in GF mice (Fig 2F) almost perfectly counterbalances their corresponding higher energy excretion in feces (Fig 2G), such that all mice extract around 9 kcal per day from their food (Fig 2H). This is consistent with our measurements of daily energy expenditure by indirect calorimetry (Fig 2B), although it fails to explain the observed adiposity in the OligoMM12 mice (Fig 1G). Unexpectedly, the efficiency of release of calories from chow remains similar between GF and OligoMM12 mice. The gut content of both OligoMM12 and SPF mice is densely colonized, and the fecal energy density is similar. Therefore, it seems that the lower percentage of energy extracted from the food by the OligoMM12 may be less related to a poorer digestive capacity of the gnotobiotic gut microbes and more to the bioavailability of microbiota-released calories for the mouse (Fig 2I).

We therefore concluded that daily energy expenditure and daily energy absorption from food vary only within the range of experimental error intrinsic to indirect calorimetry experiments. At a fundamental level, food intake therefore seems to be well regulated by microbiota-released calories. Despite this, OligoMM12 mice have an elevated fat mass. It remains a distinct possibility that gain of fat mass depends on the cumulative effect of very small differences in energy intake and energy expenditure that are simply not resolvable in our system. An alternative explanation is that microbiota composition influences energy storage. In order to gain a deeper insight into underlying mechanisms, we carried out a series of more detailed analyses of metabolism.

## Circadian changes in RER and microbiota-derived hydrogen and short-chain fatty acids (SCFAs)

Respiratory exchange ratio (RER; the ratio of $CO_2$ produced per $O_2$ consumed) is widely used as an informative proxy for substrate utilization (i.e., glucose or fatty acids) for oxidation in tissues. We observed that GF mice have a lower RER compared to SPF mice in both light and dark phases, indicative of increased fat/decreased glucose metabolism in GF mice (Fig 3A). Intriguingly, OligoMM12 mice show circadian dependence in recovery of SPF-like metabolism, phenocopying GF mice during the light phase, and SPF mice during the dark phase (Fig 3A). These changes in RER are not related to differences in feeding patterns as all mice have a

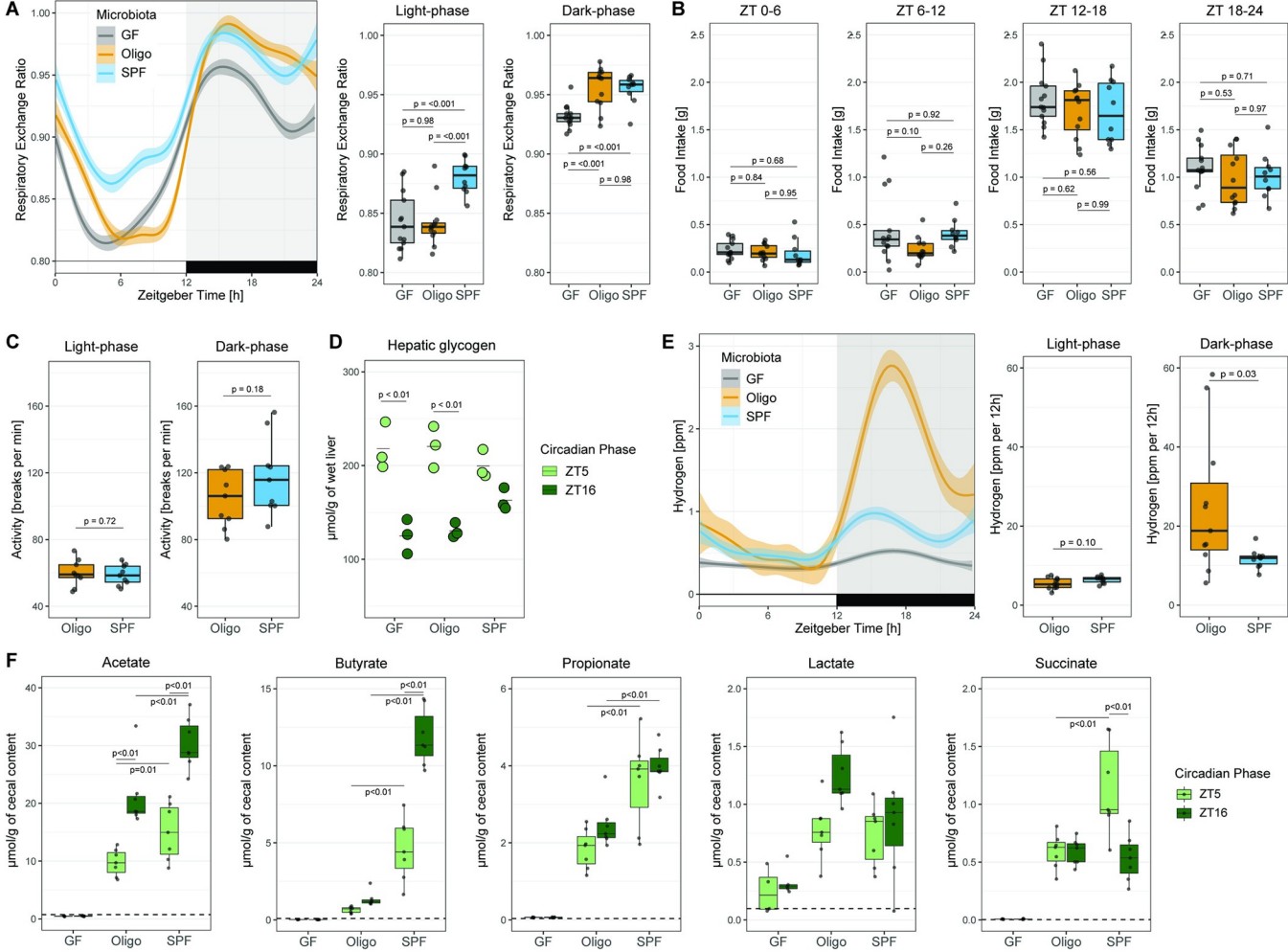

**Fig 3. Circadian changes in RER, microbiota-derived hydrogen, and SCFAs.** (A) Comparison of circadian changes in RER among GF, OligoMM12, and SPF C57B6/J mice. RER curves obtained by smoothing function of data obtained every 24 min per mouse over 10 d. Mean RER during the light phase (Zeitgeber 0–12) and dark phase (Zeitgeber 12–24). (B) Cumulative food intake during described ZT periods. Mice included in this analysis are those that underwent long-term indirect calorimetry, and they are a subset of the mice represented in Fig 2F. (C) Locomotor activity, average light phase and dark phase breaks/minute daily. (D) Hepatic glycogen and triglyceride concentration in samples obtained at Zeitgeber 5 and 16 ($N$ = 3 per group). (E) Hydrogen production, curves obtained by smoothing function of data obtained every 24 min per mouse. Area-under-curve after regression-based normalization by cecal mass during the light and dark phase (N of mice per group: OligoMM12 = 11, SPF = 10). (F) Concentration of SCFAs (acetate, butyrate, propionate) and intermediate metabolites (lactate, succinate) products in cecal content during the light phase (ZT5: GF = 4, OligoMM12 = 7, SPF = 7 mice) and dark phase (ZT16: GF = 5, OligoMM12 = 7, SPF = 7 mice). Number of mice per group in all figures unless otherwise specified: GF = 13, OligoMM12 = 12, SPF = 10. $p$-values obtained by Tukey's honest significance test. Data underlying this figure are supplied in S1 Data. GF, germ-free; RER, respiratory exchange ratio; SCFA, short-chain fatty acid; SPF, specific-opportunistic-pathogen-free; ZT, Zeitgeber time.

similar food intake pattern during the periods in which their RERs differ the most (Fig 3B). Another critical determinant of RER is locomotion. Unfortunately, we did not have a system available to track locomotion within isolators. Therefore, we could not carry out reasonable locomotion analyses of GF mice without contamination. However, the OligoMM12 microbiota is sufficiently stable to work with in standard housing for short periods of time. We therefore compared locomotion activity in a standard TSE PhenoMaster system for OligoMM12 and SPF mice. This revealed no major changes in locomotion between the 2 groups at any phase of the circadian cycle (Figs 3C and S5A and S5B).

Differences in RER provided a clue that there could be differences in energy storage in mice with different microbiota status. Microbial fermentation products, including SCFAs and lactate, can be directly used as energy and carbon sources by the murine host and are generated by the microbiota via processes that liberate molecular hydrogen. We therefore quantified hepatic concentrations of glycogen, and cecal concentrations SCFA, at ZT5 (5 h into the light phase) and ZT16 (4 h into the dark phase). Hydrogen was measured continuously during the circadian cycle.

Hepatic glycogen levels show a circadian rhythm, which usually peaks early during the transition between dark to light phase (ZT2 to 4) and drops to its minimum during the early hours of the dark phase (ZT14 to 16) in nocturnal rodents [48,49]. We found similar accumulation of hepatic glycogen in GF, OligoMM12, and SPF mice at ZT5; however, GF and OligoMM12 liver glycogen levels drop lower than SPF mice at ZT16 (Fig 3D), potentially consistent with more rapid exhaustion of hepatic glycogen supplies.

Hydrogen, a by-product of fiber fermentation by the microbiota, was also measured in the exhaust air of the metabolic cages. We found a clear circadian pattern in hydrogen production in OligoMM12 and SPF mice (Fig 3E). Hydrogen levels in OligoMM12 and SPF mice decreased down to the limit of blank (GF level as reference) during the light phase, to later peak after food intake resumes during the dark phase. In addition, OligoMM12 mice showed a higher production of hydrogen than SPF mice during the dark phase even after regression-based normalization by cecal mass (Fig 3E), i.e., the OligoMM12 microbiota produced hydrogen at a higher rate per cecal content mass than the SPF microbiota. Notably, this circadian rhythm of hydrogen production was not associated with changes either in community composition or bacterial load of the cecal microbiota in OligoMM12 mice (S6 Fig), but rather with altered metabolic activity of the bacteria present.

SCFA are the other major output of bacterial fermentation in the large intestine, as well as being key bioactive compounds produced by the large intestinal microbiota. SPF mice showed the highest cecal concentrations of acetate, butyrate, and propionate during both the light phase and dark phase, indicating efficient fermentation (Fig 3F). Interestingly, OligoMM12 mice showed only 20% to 50% of the SCFA concentrations observed in SPF mice, but instead showed high production of lactate during the dark phase (Fig 3F). In GF mice, all analyzed metabolites had levels below the limit of the blank except for lactate, which could correspond to host-produced L-lactate [50] (our assay is not able to differentiate the enantiomers). As the total mass of cecum content is widely different among GF, OligoMM12, and SPF mice, we also estimated the total quantity of each compound in the cecal content by multiplying the concentration (Fig 3F) by the cecal mass for each group (Fig 1C) while propagating the uncertainty of each measurement. This transformation has quite a major impact on how these data can be interpreted: When taking cecal mass into account, OligoMM12 mice have considerably higher levels of acetate during the light and dark phase and of propionate during the dark phase than SPF mice, while butyrate levels remain low. There is also an increased abundance of lactate and succinate in the OligoMM12 cecum content (S5C Fig). Although we cannot directly link these microbial metabolites to the phenotype of the OligoMM12 mice, this underlines the

major differences in microbial metabolite profiles in the large intestine when comparing GF, gnotobiotic, and SPF mice. High lactate production by the microbiome certainly warrants further study for potential effects on the host.

## Circadian changes in liver and plasma metabolites in GF, OligoMM12, and SPF mice

Finally, to increase our metabolic resolution, we applied ultraperformance liquid chromatography coupled with mass spectrometry (UPLC/MS) to perform untargeted metabolomics in the liver and plasma during the light (Zeitgeber 5) and dark phase (Zeitgeber 16) in GF, OligoMM12, and SPF mice. Correlating to what we observed in the RER during the light phase, GF and OligoMM12 cluster closely and are clearly separated from the SPF in the light phase of principal component analysis for both liver and plasma samples (Fig 4A). However, no major shift towards the SPF metabolome was seen during the dark phase for OligoMM12 liver and plasma samples (Fig 4B). Therefore, although RER and glycogen levels clearly show GF-like patterns during the light phase and SPF-like patterns during the dark phase, the underlying metabolome circadian shifts attributable to the microbiome in OligoMM12 mice are subtle and generally closer to GF signatures than to SPF signatures in both liver and plasma samples.

We used the package MetaboAnalystR [51] to identify putative compounds that are significantly different in pair comparisons between OligoMM12 mice and their GF and SPF counterparts by untargeted peak extraction. These were then mapped onto metabolic pathways using the KEGG database. We found several pathways differentially enriched when OligoMM12 mice were compared to GF or SPF counterparts during the light and dark phase in liver (Fig 4C and 4E) and plasma (Fig 4D and 4F), including butanoate metabolism, amino acid biosynthesis and degradation, primary bile acids production, and fatty acid metabolism. Additionally, we selected compounds that belong to these differentially enriched pathways or have been previously identified to have circadian changes in obese patients [52], confirmed their structure using chemical standards, and performed a targeted peak extraction for a more precise comparison among groups (S7 and S8 Figs; full list of compounds in S1 Table). We observed that OligoMM12 show a different pattern when compared to GF or SPF mice depending on the compound analyzed, the site (live or plasma), and the circadian phase. For example, the ketone body β-hydroxybutanoate (which is the conjugated form of β-hydroxybutyrate and part of the butanoate metabolism pathway) is higher in the plasma of the OligoMM12 mice during both light and dark phase. For other compounds such as certain amino acids, and depending on the circadian phase and site, OligoMM12 have a similar pattern to GF (i.e., leucine) or SPF (i.e., L-glutamate and glycine). Finally, for many of these metabolites, the OligoMM12 microbiota produce an intermediate phenotype between GF and SPF mice, as in the case of the bile acids β-murocholate.

As bile acid profiles have been previously linked to increased fat mass, we also extracted all nonambiguous data relating to bile acids from our UPLC/MS data. This shows a good agreement with published literature on this topic (for example, elevated β-murocholate and Tauro-β-murocholate in the liver bile acid pool of GF mice, when compared to colonized animals [11,53]). The circadian rhythm dependence varies between bile acid examined, tissue examined, and microbiota status generating a complex picture that warrants deeper exploration.

## Discussion

Since the early days of nutritional studies, there has been a clear interest to understand the role of microbiota in host morphology, physiology, and nutrition [54,55]. Pioneering work comparing GF rats with conventionally raised counterparts already described differences in food

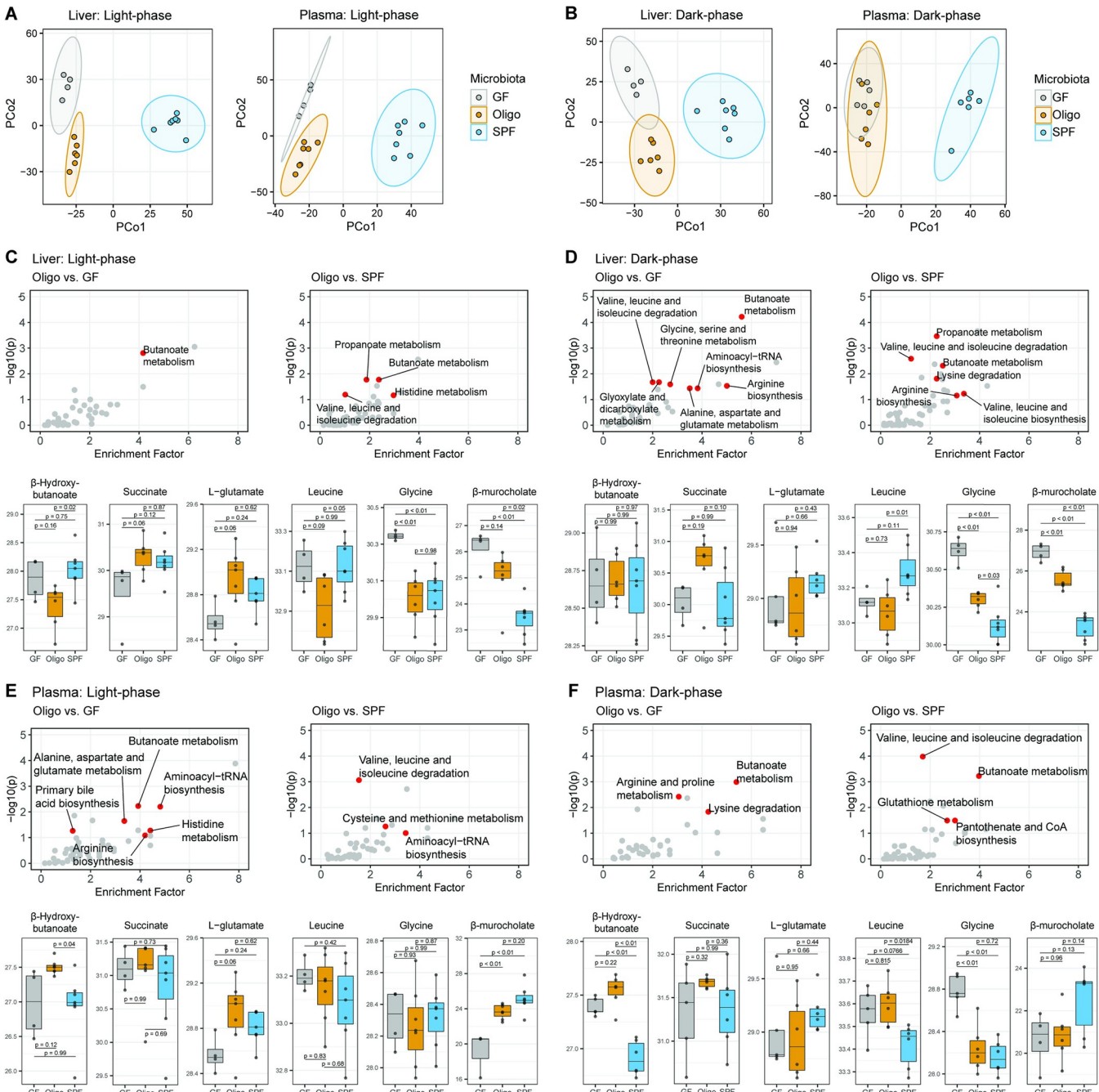

**Fig 4. Metabolic profile comparison of GF, OligoMM12, and SPF C57B6/J mice by UPLC/MS.** (A and B) Principal coordinate analysis using Canberra distances of metabolites identified by untargeted UPLC/MS in liver and plasma during the (A) light phase (Zeitgeber 5) and (B) dark phase (Zeitgeber 16). (C-F) Metabolic pathways identified in the KEGG PATHWAY database; red dots represent pathways containing compounds differentially enriched in OligoMM12 vs.GF and OligoMM12 vs. SPF comparisons and selected compounds obtained by targeted peak extraction from differentially expressed pathways. Samples obtained during the light phase (Zeitgeber 5) and dark phase (Zeitgeber 16) in (C and D) liver and (E and F) plasma. *p*-values obtained by Tukey's honest significance test after log$_2$ transformation of area value. Number of mice per group: Liver ZT5: GF = 4, OligoMM12 = 6, SPF = 7; ZT16: GF = 4, OligoMM12 = 6, SPF = 7 / Plasma ZT5: GF = 4, OligoMM12 = 7, SPF = 7; ZT16: GF = 5, OligoMM12 = 6, SPF = 6. Data underlying this figure are supplied in S1 Data. GF, germ-free; SPF, specific-opportunistic-pathogen-free; UPLC/MS, ultraperformance liquid chromatography coupled with mass spectrometry; ZT, Zeitgeber time.

intake, energy extraction from diet, and energy expenditure by indirect calorimetry [44,56]. More recently, researchers have explored the effect of specific complex microbiota communities and how they influence energy metabolism and body composition in the host [9,57,58]. Here, we extend and clarify some of these observations via use of a well-established gnotobiotic mouse model consisting of 12 cultivable microbiota strains and a custom-built isolator-housed metabolic cage system that permits longitudinal analysis of GF and gnotobiotic animals.

By carefully checking the validity of different measurement types, we found no significant difference in lean body mass among GF, gnotobiotic (OligoMM12), and conventionally raised (SPF) mice. Although lean mass represented a lower percentage of the total mass in GF mice, this was mainly attributable to increased cecal water retention in these animals. Interestingly, there was a significant increase in fat depots in OligoMM12 mice compared to GF and SPF animals. Previous studies have also found increased fat depots during conventional/low-fat diet feeding in mice colonized with a gnotobiotic microbiota community [21] or SPF [2,14,58] when compared with GF mice. Our results using fat depot dissection showed only a very weak trend for white adipose tissues between GF and SPF mice, which may be attributable to differences in housing (temperature, cage-type, chow composition) and colony (genetic background, SPF microbiota composition, age). It has been shown that GF mice transplanted with microbiota derived from obese donors accumulated more fat mass compared to those transplanted with microbiota derived from lean donors [9,36,57], with correlates identified to individual species/strain abundance [59,60]. SPF microbiota matching more closely to those from obese donors could therefore be expected to give differing results to ours. In contrast, minimal microbiota communities such as the OligoMM12 can be perfectly replicated across sites [24] and can help to clarify the complex processes linking microbiota and host metabolism [61]. Further exploration of the metabolic effects of the OligoMM12 microbiota community, and extended versions thereof, has potential to clarify if specific strains, species, or functional classes [62] are sufficient and necessary to drive the development of increased fat depots in these mice.

We further observed no significant difference in energy expenditure in GF, OligoMM12, and SPF. This is in line with some studies that have reported no significant difference in energy expenditure between GF and SPF mice [3,13]. These results are in contrast to other work [2,15,35,56], but the discrepancies can potentially be explained by the methods applied for normalizing energy-expenditure data. Normalization of mass-dependent variables by a per-mass (or allometric transformation) ratio has been recognized as a common source of controversy [63–65], especially with large changes in body mass composition [66,67], and there have been several publications calling for the use of better statistical methods [41,68,69]. Water and indigestible solute retention in the cecum lumen of GF and gnotobiotic mice can contribute up to 10% of the total body mass and should be considered metabolically inert. It is therefore unsurprising that when the cecal content mass is very different among groups, using total body mass for normalization introduces a considerable bias in normalized energy expenditure estimation. Interestingly, it was long ago observed that surgical removal of the cecum equalized the oxygen consumption between GF and conventional rats, as well as other measurements normalized by total body mass [35]. With normalization using linear regression models based on lean mass and fat mass [43], we and others found no significant differences in energy expenditure by indirect calorimetry between GF and SPF mice under standard chow diet conditions [3,13].

An additional important confounder that we encountered was high variability of fat mass readouts obtained by EchoMRI when comparing mice with major differences in intestinal colonization levels. This could be attributed to variable calling of the fluid-filled ceca of gnotobiotic animals as either fat or water, compared with more accurate calling in conventional mice, revealing an important limitation of these systems. Surprisingly, the EchoMRI estimate of fat

mass increased in SPF mice after abdominal dissection and cecum removal. Previous studies reported a tendency to higher values of fat mass in dead animals when compared to live [38,39], which we could replicate. However, this was a much smaller effect than cecum removal. We could not find reports of EchoMRI measurements after major anatomical changes such as cecum removal, and we cannot accurately explain this phenomenon. Therefore, we recommend caution in the use of EchoMRI for fat mass measurements in mice with marked anatomical differences (i.e., enlarged cecum) and recommend physically dissected fat mass as a more useful readout.

We are also keen to point out the more general limitations of our observations: Only 1 gnotobiotic microbiota and 1 SPF microbiota were analyzed, and our conclusions pertain exclusively to these. We in no way exclude the possibility that some microbiota constituents or conformations can influence host energy expenditure [36] and/or body composition [9,57,70]. In addition, it should be noted that indirect calorimetry is an inherently noisy data type, and small differences in daily energy expenditure are impossible to resolve via this technique [69,71].

Nevertheless, the lack of measurable difference in energy expenditure between GF, OligoMM12, and SPF mice is aligned with our finding that the amount of energy obtained by ad libitum food intake was also remarkably similar among the groups. GF mice seem to accurately compensate the lower capacity of energy extraction from diet by increasing food intake. While this seems generally to be in agreement with models that described the regulation of appetite (and therefore energy intake) by the basal energy requirement of the individual [72,73], it remains surprising given the discrepancy in the types of substrates available for oxidative metabolism in colonized and GF mice, revealed by RER differences. Although GF mice have a longer total gastrointestinal transit time than SPF mice [74], very little calorie absorption from food can occur after ingested food reaches the cecum of a GF mouse, while an SPF mouse will release usable energy from their food via microbial fermentation for several more hours in the cecum and colon, generating a major time difference in the absorption of calories after eating in GF and SPF animals. This compensation seems also to function in mice colonized with the OligoMM12 microbiota, where despite robust microbial fermentation (read out as hydrogen and fermentation product production) and identical fecal energy density to SPF mice, energy recovery from ingested food is poor due to the volume of feces shed. A clear conclusion from these observations is that microbiota-dependent changes in metabolic substrates, and timing of calorie absorption, are well integrated in the murine central regulation of appetite over the course of a day [75].

Interestingly, energy density of dry feces in GF mice was lower compared to OligoMM12 and SPF mice. Previous results have found a similar difference (approximately 0.1 kcal/g) when comparing GF and SPF rats under standard chow [44]. We theorized that this difference is due the contribution of energy stored in bacterial mass, which we estimated is in the range of 0.5 kcal/g per gram of feces. However, this observed difference in the fecal caloric content seems to depend on the type of diet, as GF and SPF mice under a high-fat diet showed a similar caloric content [4]. In addition, caloric uptake by the microbiota may be dependent on specific microbiota composition. Although we did not observe a difference in the fecal energy content among OligoMM12 and SPF mice, previous studies have shown that particular microbiota compositions allow more energy to be lost in the fecal output [9].

Despite this broadly successful regulation of food intake and energy expenditure, at the molecular level, major differences were observed between the mice with different microbiota. First, OligoMM12 mice displayed an RER at the GF level during the light phase (when mice typically sleep and fast) but raised up to SPF levels during the dark phase (i.e., when mice are active and eating). It therefore appears that the OligoMM12 microbiota better recapitulates the

microbiome effects on the host energy substrate use during the dark (active) phase when food-derived carbohydrates are abundant in the large intestine, but not in the light (sleeping) phase when mainly host-derived carbon sources are available in the large intestine. We could directly exclude food intake and locomotion as major drivers of this altered RER. Interestingly, SPF had a higher RER than OligoMM12 and GF mice during the light phase despite no difference in the levels of hepatic glycogen at the beginning of this phase. This indicates that GF and OligoMM12 are using more fatty acids, and potentially that SPF mice have more prolonged access to carbohydrate substrates produced by their more complex microbiota or stored in other body sites. Improved carbon release from dietary fiber by the SPF microbiota would also be in line with a predominance of succinate and lactate in the OligoMM12 cecum, at the expense of propionate and butyrate that are more abundant in the SPF cecum. In complex microbiotas, lactate is typically further metabolized to butyrate by specific firmicutes [76–78], which may be lacking or insufficiently abundant in the OligoMM12 mice. As lactate can inhibit lipolysis in adipocytes [79,80], this raises an interesting theme for follow-up studies to define the role of microbiota-derived lactate in host metabolism. Partially in line with the RER data, we also observed that the liver and plasma metabolite profiles of OligoMM12 mice clustered closer to GF mice than to SPF mice. Although a small shift in the liver metabolome could be observed in the OligoMM12 liver during the dark phase, this clearly demonstrates major metabolic effects of a complete microbiota that are not reconstituted by the OligoMM12 strains. In addition, certain amino acids were differentially represented between OligoMM12 and GF or SPF mice, as it has been described previously [81,82]. Interestingly, OligoMM12 had a bile acid profile closer to GF than SPF mice, for example, showing GF levels of hepatic β-murocholic acid and taurine-β-murocholic acid, the predominant bile acid in the liver of GF mice [11]. Follow-up studies with manipulation of the OligoMM12 microbiota or metabolic interventions are a promising tool to pull apart the circadian effects on RER, the influence of an unusual fermentation product profile, and other more subtle metabolic changes on overall metabolic health of the murine host.

In conclusion, our study showed that isolator-based indirect calorimetry is possible and allows detailed analysis of the metabolism of GF and gnotobiotic mice in real time. Data generated with this system demonstrated that microbiota-released calories are well integrated in host energy balance and that daily energy expenditure was not significantly influenced by microbiota composition in our mice. Nevertheless, mice colonized with the OligoMM12 gnotobiotic microbiota accumulated more fat mass and display a GF-like RER during the light phase but an SPF-like RER during the dark phase, indicative of altered metabolic substrate usage and energy storage. Correspondingly, the liver metabolome of mice colonized with the OligoMM12 showed alterations in bile acid, fatty acid, and amino acid metabolism, despite overall clustering with the GF liver metabolome. This reveals the potential for gnotobiotic microbiota communities to investigate the mechanisms underlying the influence of microbiota on host metabolic health. As microbial dysbiosis is associated with a range of human diseases, circadian analysis of energy balance represents a crucial tool in the mining of microbiome data for therapeutic and diagnostic purposes.

## Methods

### Animals

We used C57B6/J male mice aged between 12 to 14 weeks. We compare GF, with a 12-strain gnotobiotic microbiota [23] (OligoMM12), and SPF mice. The OligoMM12 mice used in this study were colonized from birth as they belonged to a colony, originally established by colonizing GF mice with 12 bacterial strains and later checking their engraftment by qPCR [24]. GF

and OligoMM12 mouse lines are bred and maintained in open-top cages within flexible-film isolators, supplied with HEPA-filtered air, and autoclaved food and water ad libitum. As we are aware that housing conditions may influence behavior and potentially metabolism, we also bred and maintained a SPF colony under identical conditions inside a flexible-film isolator specifically for this study, such that all mice experienced identical living conditions, food, and water. Mice were adapted for between 24 to 36 h after transfer from the breeding isolators to the isolator-based metabolic chambers. For long-term experiments, mice were periodically rehoused in couples for short periods of times to avoid stress of extended single-housing conditions. In all cases, animals were maintained with standard chow (diet 3807, Kliba-Nafag, Switzerland) and autoclaved water. GF status was confirmed at the end of the long-term experiments by culturing cecal content in sterile BHIS and YPD media in aerobic and anaerobic conditions for a week. In addition, cecal content was frozen at −20˚C for a week, then stained with SYBR Gold and assessed by bacterial flow cytometry [83] using similarly processed SPF mice cecal content as positive control for the presence of bacteria. All experiments were conducted in accordance with the ethical permission of the Zürich Cantonal Authority under the license number ZH120/19.

## Indirect calorimetry

The isolator-housed TSE PhenoMaster system allows instantaneous measurements of oxygen, carbon dioxide, and hydrogen levels as well as total feed and water consumption while keeping a strict hygiene level of control. The metabolic isolator system consists of an adapted set of 2 flexible-film surgical isolators, each of them housing 4 metabolic cages from the TSE PhenoMaster system (TSE Systems, Germany). Room air is pulled into the isolator by a vacuum pump passing through a double set of HEPA filters. Then, each cage is connected via a second HEPA filter through the back wall of the isolator to the CaloSys setup, which pulls sterile air from the isolator into the cages using negative pressure. Air coming from the cages is dehumidified at 4˚C and sequentially passed by a Sensepoint XCD Hydrogen gas analyzer (Honeywell Analytics, Hegnau, Switzerland) and standard oxygen and carbon dioxide censors provided in the TSE PhenoMaster system. A 2-point calibration of all analyzers using reference gases was performed within 24 h before each animal experiment. Data were recorded using a customized version of the TSE PhenoMaster software modified to integrate hydrogen measurements.

For indirect calorimetry measurements, the animals were transported in pre-autoclaved, sealed transport cages from the breeding isolators into the metabolic isolator system. Mice were single housed and adapted for between 24 to 36 h before starting recording measurements to ensure proper access to food and water as well as account for initial exploratory behavior. Mice were kept up to 10 d at a stable temperature (21 to 22˚C) with ad libitum availability of standard chow and water. The days were divided into a dark and light period of 12 h each. In this study, we kept the air flow of 0.4 L/min and recorded individual cage data (gases production and food/water consumption) every 24 min (time set per cage for measurement stabilization 2.5 min). In long experiments, mice were periodically pair-housed for 24 h to prevent stress due to prolonged single housing.

## Body composition measurements

At the end of the experiment, mice were fasted for 4 h (Zeitgeber 1 till 5) before for body composition measurements. We used magnetic resonance whole-body composition analyzer (EchoMRI, Houston, USA) to analyze mice body composition (lean and fat mass). Then, mice were killed using $CO_2$ according to approved protocols. Total body mass was obtained by

weighing the full carcass, and cecum was dissected and weighed by 1 investigator (DH). For a set of mice, we remeasured body composition by EchoMRI after cecum removal and compare it with the composition observed in live animals (S2 Fig). Finally, fat depots were dissected from all mice after cecum removal by another investigator (WS) that was blinded to the hygiene status and cecum size of the mice. iBAT, iWAT, and vWAT were sampled and weighted. For a group of SPF mice, body composition by EchoMRI was performed also before cecum removal (S2K Fig).

## Food intake, fecal samples, and bomb calorimetry

Daily food intake was obtained as the mean value of food intake recorded by the TSE Pheno-Master system during the course of the experiment. In addition to the mice reported in the indirect calorimetry experiments, we also collected food intake data from a set of selected experiments in which we collected fecal pellets produced during 24 h. For daily fecal excretion measurements, we cleaned up the bedding in the cage and replaced it for a clean and reduced amount of bedding. After 24 h, we collected the mix of bedding and fecal pellets. Fecal pellets were manually collected from the bedding, transferred to 15 ml tubes and stored at −20˚C until bomb calorimetry. Before bomb calorimetry, fecal samples were freeze dried in a lyophi-lizer overnight (ALPHA 2–4 LDplus, Christ, Germany) and dry mass recorded. We used a C1 static jacket oxygen bomb calorimeter (IKA, Germany) to quantify the residual energy present in these dry fecal pellets, using approximately 0.2 to 0.5 g of material. Energy content was nor-malized to grams of dry fecal pellets.

## Locomotor activity measurements

OligoMM12 and SPF mice were transferred to a different facility and single-housed in a con-ventional TSE PhenoMaster equipped with ActiMot3 Activity module for locomotor activity measurement. After 1 d of adaption, standard indirect calorimetry plus locomotor activity was recorded every 20 min for the next 5 d. Locomotor activity was reported as the average light beam breaks (XT+YT) per min.

## Sample obtention and preparation for metabolomics, and 16S sequencing

Approximately at Zeitgeber 5 and 16, mice of each group were killed, and liver and plasma samples collected. To minimize variations among mice, individual mice were killed with $CO_2$ and sampled as fast as possible. Blood was obtained by cardiac puncture, collected in lithium heparin coated tubes, and kept on ice for further processing. Mice were perfused with PBS and liver samples were obtained by dissection of the lower right lobe, collected on a 2-ml Eppen-dorf tube and flash frozen in liquid nitrogen. Finally, between 60 to 80 mg of cecal content was collected in a 2-ml Eppendorf tube and flash frozen in liquid nitrogen. After all samples were obtained, blood samples were centrifuged 8,000 rcf for 5 min, supernatant collected, and flash frozen in liquid nitrogen. Samples were kept at −80˚C until further processing.

## Metabolomics by UPLC/MS

**Short-chain fatty acid quantification by UPLC/MS.** Samples were first homogenized in 70% isopropanol (1 mL per 10 mg sample), centrifuged. Supernatants were used for SCFA quantification using a protocol similar to previously described [84]. Briefly, a 7-point calibra-tion curve was prepared. Calibrators and samples were spiked with mixture of isotope-labeled internal standards, derivatized to 3-nitrophenylhydrazones, and the derivatization reaction was quenched by mixing with 0.1% formic acid. Approximately 4 μL of the reaction mixture

was then injected into a UPLC/MS system, [M-H]– peaks of the derivatized SCFAs were fragmented, and characteristic MS2 peaks were used for quantification.

**Untargeted UPLC/MS.** Samples were thawed on ice. Serum samples were diluted with 90% methanol in water with a volumetric ratio of 1:7, incubated for 10 min on ice for allowing protein to precipitate. Liver samples were mixed with 75% methanol in water (2 mL/100 mg liver), homogenized using a Tissue Lyser (Qiagen, Germany) at 25 Hz for 5 min. The result mixtures were centrifuged at 15,800*g*, 4˚C for 15 min. Approximately 100 μL of the supernatants were filtered with 0.2 μm reversed cellulose membrane filter and transferred to sample vials and used for UPLC/MS analysis with an ACQUITY UPLC BEH AMIDE column (1.7 μm, 2.1 × 150 mm, Waters). Another 400 μL of the supernatants were then lyophilized and resuspended in 80 μL 5% methanol in water, sonicated, filtered, and used for UPLC/MS analysis with an ACQUITY UPLC BEH C18 column (1.7 μm, 2.1 × 150 mm, Waters, RP column).

An ACQUITY UPLC system (I-Class, Waters, MA, USA) coupled with an Orbitrap Q-Exactive Plus mass spectrometer (Thermo Scientific, San Jose, CA) were used for UPLC/MS analysis. For the AMIDE column a flow rate of 400 μL/min was used with a binary mixture of solvent A (water with 0.1% formic acid) and solvent B (acetonitrile with 0.1% formic acid). The gradient starts from 1% of A, then gradually increases to 70% of A within 7 min. Then a 1% of A is kept for 3 min. The column was kept at 45˚C and the autosampler at 5˚C.

For the RP column, the flow rate was set to 240 μL/min using a binary mixture of solvent A (water with 5% methanol and 0.1% formic acid) and solvent B (methanol with 0.1% formic acid). The gradient starts from 95% of A, then gradually decreases to 5% of A within 10 min. A 100% solvent of B is kept for 2 min, then a 100% of A is kept for 2 min to restore the gradient. The column was kept at 30˚C and the autosampler at 5˚C.

The MS was operated at a resolution of 140,000 at m/z = 200, with automatic gain control target of $2 \times 10^5$ and maximum injection time was set to 100 ms. The range of detection was set to m/z 50 to 750. Untargeted MS data were extracted from raw MS files by using XCMS [85] in R (v3.6.1) and then subject to pathway enrichment by using MetaboAnalystR [51]. From all identified pathways, we selected those with a −log(p) value lower than 1 and those that include at least 5 significantly different compounds with no identical molecular weight.

**Compound identification and targeted peak extraction.** Chemical standards of selected compounds were diluted to 10 μg/mL and were analyzed using the UPLC/MS methods described before. Identification was done by comparing retention time and MS2 spectra in liver/plasma samples with the chemical standards [52]. After confirming the chemical identities of the compounds, targeted peak extraction was done using Skyline (v21.1) [86].

**Quantification of bacterial density by flow cytometry.** We used cecal content of GF, OligoMM12, and SPF mice sampled as described before to quantify bacterial density by flow cytometry during the light and dark phase. Briefly, approximately 20 to 50 mg of cecum content was homogenized in 2 mL Eppendorf tubes with 1 mL of PBS, using a 2-mm metal bead in a Tissue Lyser at 25 Hz for 2.5 min. After homogenization, tubes were left on the bench for 5 min for precipitation of big food particles. A mix of SYBR Gold (1:2,000 dilution from stock in PBS) spiked with fluorescent counting beads (Fluoresbrite Multifluorescent Microspheres 1.00 μm, Polysciences, USA) was prepared at a concentration $4.55 \times 10^3$ beads/μL. Then, 2 μL of homogenized content was added to the SYBR Gold plus beads mix and incubate at room temperature for 15 min. Samples were acquired by flow cytometry as described before for 1 min [83]. GF samples were used as negative controls to set the gates for SYBR Gold-positive bacterial particles. Bacterial counts were normalized to bead counts to estimate the concentration of bacteria in the sample.

### 16S sequencing for OligoMM12 community composition analysis

**DNA extraction.** For enzymatic lysis, roughly 30 mg of flash-frozen cecum content per sample were incubated in 100 μl of 1× TE buffer (30 mM Tris-HCl and 1 mM EDTA) supplemented with 30 mg/ml Lysozyme (Sigma-Aldrich), 1.6 U/ml Proteinase K (New England Biolabs), 10 U/ml Mutanolysin (Sigma-Aldrich), and 1 U/μl SUPERase•In RNase Inhibitor (Invitrogen) at room temperature for 10 min. To aid disruption, one 2-mm metal bead was added, and the samples were vortexed every 2 min. Subsequently, the samples were mixed with 550 μl RLT Plus buffer (Qiagen) complemented with 5.5 μl 2-beta-mercaptoethanol (Sigma-Aldrich) and prefilled tubes with 100 μm Zirconium beads (OPS Diagnostics LLC). The samples were disrupted twice at 30 Hz for 3 min using the mixer mill Retsch MM400 with 5-min incubation at room temperature between each disruption.

DNA was extracted from all samples with the DNA/RNA Mini kit (Qiagen) following the standard protocol and eluting the DNA in 100 μl elution buffer (EB). One water sample was used as a negative extraction control and subsequently split into 3 negative library controls undergoing the same library preparation as all samples. The integrity and quality of the extracted DNA was assessed on a Qubit and Fragment Analyzer respectively. The DNA was purified by overnight ethanol precipitation at −20°C in 275 μl ice-cold Ethanol (Sigma-Aldrich), 10 μl 3 M Sodium acetate (Invitrogen), and 1 μl 20 mg/ml Glycogen (Invitrogen) with subsequent centrifugation at 4°C for 30 min and 2 wash steps in 500 μl ice-cold 75% Ethanol with centrifugation at 4°C for 10 min each time. The DNA purity was assessed on a Nanodrop.

**Sequencing for community composition analysis.** 16S amplicon libraries were generated from 50 ng input DNA with the Illumina primer set 515F Parada [87] and 806R Apprill [88], 12 cycles in PCR 1 and 13 cycles in PCR 2. Three positive controls containing 11 ng input DNA of ZymoBIOMICS Microbial Community DNA Standard II (Zymo Research, Germany) were used. Illumina Unique Dual Indexing Primers (UDP) were used for library multiplexing. A 12-pM library pool spiked with 20% PhiX was sequenced at the Functional Genomics Center Zurich using the MiSeq platform and 2 × 300 bp PE-reads with a target fragment size of 450 bp resulting in approximately 400,000 reads per sample. One sample was excluded from the analyses due to missing sequencing reads.

## Data analysis

**Data quality control.** To facilitate analysis across different experimental runs, all times were converted into ZT (h), where 0 to 12 represents the light phase and 12 to 24 represents the dark phase. Any datapoint taken before the start of the first occurrence of ZT = 0 was discarded. To account for faulty measurements caused by measurement imprecision, equipment malfunction or other disruptive events, datapoints were removed from the raw datasets according to criteria based on statistical and biological arguments. Food consumption values of 0.01 g during the 24-min intervals were considered as measurement noise and discarded. Negative values for food and water consumption, as well as oxygen ($dO_2$) and carbon dioxide ($dCO_2$) differentials between the measurement chambers and the reference chamber were also considered as measurement noise and discarded. For the remaining subsets of measurements from the individual mice, we cleaned up outlier measurements in food and water intake by eliminating values greater than 75th percentile + 1.5 times interquartile range. Potential sources for outlier measurements in food and water consumption observed included leaky water bottles and loss of food pellets during mice husbandry procedures. A similar approach was used to eliminate outliers from $dO_2$ and $dCO_2$ values below 25th percentile − 1.5 times interquartile range. Potential sources for outlier measurements in gas differentials included

inappropriate sealing of individual metabolic cages or clogging of pre-analyzer filters. Oxygen consumption ($VO_2$) and $CO_2$ production ($VCO_2$) was calculated using $dO_2$ and $dCO_2$ and the Haldane transformation as described before [68]. Energy expenditure was estimated from $dO_2$ and $dCO_2$ using Weir's approximation [89]. As one of the study objectives is to explore circadian patterns, if more than 20% of datapoints had to be removed from a particular day for a particular mouse, all other datapoints from that subset were discarded as well. After the cleanup process described above, the data from all different experiment runs were pooled together for further analysis. The above processes lead to a reduction in dataset size from 10,472 to 9,453 entries.

In the targeted and untargeted metabolomic analysis, some samples were excluded from further analysis due to technical reasons. Liver and plasma samples from 1 animal (L934 and P934) were excluded due to altered phenotype observed during sample acquisition. Additionally, 1 plasma sample (P939) and 2 liver samples (L914 and L930) were excluded due to errors in dilutions during sample preparation.

**16S amplicon analysis for OligoMM12 community composition.** Raw sequencing reads from all samples and 3 positive/negative controls served as input for the inference of ASVs using dada2 v1.22 [90]. Primer sequences (515F, 806R) were removed using cutadapt v2.8 [91], and only inserts that contained both primers and were at least 75 bases were kept for downstream analysis. Next, reads were quality filtered using the filterAndTrim function of the dada2 R package (maxEE = 2, truncQ = 3, trimRight = (40, 40)). The learnErrors and dada functions were used to calculate sample inference using pool = pseudo as parameter. Reads were merged using the mergePairs function and bimeras were removed with the removeBimeraDenovo (method = pooled). Remaining ASVs were then taxonomically annotated using the IDTAXA classifier [92] in combination with the Silva v138 database [93] available at http://www2.decipher.codes/Downloads.html. The resulting ASV table was used to check for contaminations with the decontam R package [94] using both frequency-based and prevalence-based classification with a single probability threshold of 0.05 computed by combining both probabilities with Fisher's method (method = combined). ASVs classified as contaminants as well as the positive/negative controls were excluded from downstream analyses. The remaining ASV abundance table was downsampled to a common sequencing depth (approximately 130,000 reads per sample) to correct for differences in sequencing depth between samples using the rrarefy function of the vegan R package.

Relative abundance plots for the light and dark phase time points were generated separately. The OligoMM12 strains were identified using the package bio for rRNA sequence extraction from the Genbank accessions described earlier [23] and the tool VSEARCH (search_exact) for sequence alignment to the 16S sequences from the detected ASVs. ASVs with a maximum relative abundance below 0.05% across all samples were grouped into "Other". *Megasphaera* was detected at the genus level at a mean relative abundance of 0.06% but was also grouped into the category "Other" since it was not knowingly part of the original OligoMM12 community. The category "Other" in total amounted to roughly 0.11% of the total relative abundances, thus the oligo strains represented at least 99.8% of the detected ASV abundances.

## Statistical analysis

From the resulting dataset, energy expenditure over a certain period was calculated as the area under the curve (trapezoid interpolation) of instantaneous values obtained during the 24-min measurements intervals. Food intake values calculated over a certain time are always cumulative. To compare different mice in the above variables, variations in body mass and composition between individuals need to be accounted for. As suggested in several publications

[41,42,69], this was done by regression-based analysis of covariance (ANCOVA). As such, a linear regression is performed on energy expenditure as a function of lean body mass and fat depots mass, with the microbiota group as a qualitative covariate. Then, each individual value is replaced by the sum of the corresponding residual and the energy expenditure predicted by the linear model using the average lean body and fat depot mass (calculated over all groups). Hydrogen production (difference in hydrogen concentration between the measurement chambers and the reference chamber) was adjusted in analogous fashion, using cecal mass (as a proxy for total gut microbiota mass) as a predictor.

For variables where the continuous evolution during the circadian cycle is of interest (RER, gross hydrogen production), values were averaged at each time point for each individual. A generalized additive model was used to fit a smooth line to these averages using a cubic penalized regression spline (using R function mgcv::gam with formula y ~ s(x; bs = "cs")).

For estimating derived variables (i.e., daily energy excretion), we used the R package "errors" [95]. This package links uncertainty metadata to quantity values (i.e., mean "daily fecal dry mass excretion", mean "fecal energy content"), and this uncertainty is automatically propagated when calculating derived variables (i.e., "daily energy excretion" = "daily fecal dry mass excretion" × "fecal energy content"). Uncertainty is treated as coming from Gaussian and linear sources and propagates them using the first-order Taylor series method for propagation of uncertainty.

For the principal coordinate analysis, we used the *pcovar* function included in the R package "dave" for calculating Canberra distances among metabolites.

All group comparisons were analyzed by ANOVA and Tukey's honest significance test. For comparisons of metabolites identified by targeted peak extraction among groups, area values were $\log_2$ transformed before the statistical test.

### Resource availability

**Materials availability.**   This study did not generate new unique reagents.

### Supporting information

**S1 Fig. Sterility test in isolator-based indirect calorimetry system.** (A) OD measurement of BHIS liquid cultures incubated overnight in aerobic and anaerobic conditions. (B-C) Representative (B) BHI-blood and (C) YPD plates streaked with GF and SPF cecum content and incubated for 3 d. (D) Representative histograms bacteria flow cytometry plots of PBS, GF, and SPF cecum content stained with SYBR Gold. Data underlying this figure are supplied in S1 Data. GF, germ-free; OD, optical density; SPF, specific-opportunistic-pathogen-free. (TIF)

**S2 Fig. Cecal mass interferes with fat mass estimation by EchoMRI.** (A) Cecal mass (tissue including luminal content) as percentage of total body mass (N of mice per group: GF = 16, OligoMM12 = 12, SPF = 11) (B) Percentage of lean body mass before cecum removal. (C) Lean body mass estimated by EchoMRI with and without cecum. Measurements were taken on live animals (x-axis) and dead animals after cecum dissection (y-axis). Equations show simple linear regression for estimating lean mass without cecum based on lean mass with cecum; in brackets adjusted R-squared. (D) Lean mass difference after cecum removal. (E) Lean mass difference after cecum removal as percentage of lean mass before cecum removal. (F) Fat body mass acquired by EchoMRI before cecum removal. (G) Percentage of fat body mass before cecum removal. (H) Fat body mass estimated by EchoMRI with and without cecum. Measurements were taken on live animals (x-axis) and dead animals after cecum dissection (y-axis).

Equations show simple linear regression for estimating fat mass without cecum based on fat mass with cecum; in brackets adjusted R-squared. (I) Fat mass difference after cecum removal. (J) Fat mass difference after cecum removal as percentage of lean mass before cecum removal. (K) Fat mass measured by EchoMRI in live, dead, and cecum-removed SPF mice ($n = 9$). Number of mice per group in all figures unless otherwise specified: GF = 13, OligoMM12 = 11, SPF = 15. $p$-values obtained by Tukey's honest significance test. Data underlying this figure are supplied in S1 Data. GF, germ-free; SPF, specific-opportunistic-pathogen-free. (TIF)

**S3 Fig. Cecal mass interferes with normalization of energy expenditure.** (A-B) Comparison of circadian changes in energy expenditure (without normalization) among GF, OligoMM12, and SPF C57B6/J mice. (A) Circadian variation in average energy expenditure per time point and (B) overlayed curves obtained by smoothing function of data obtained every 24 min per mouse over 10 d. (C-E) Energy expenditure values obtained by "classical" ratio-based normalization methods (dividing energy expenditure values per phase by mass). (C) Area-under-curve after normalization by total mass after cecal dissection. (D) Area-under-curve after normalization by lean body mass (EchoMRI). (E) Area-under-curve after normalization by total body mass before cecal dissection. Number of mice per group in all figures unless otherwise specified: GF = 9, OligoMM12 = 8, SPF = 10. $p$-values obtained by Tukey's honest significance test. Data underlying this figure are supplied in S1 Data. GF, germ-free; SPF, specific-opportunistic-pathogen-free. (TIF)

**S4 Fig. Bacterial density in cecum content of OligoMM12 and SPF mice.** (A) Bacterial density in cecum content of OligoMM12 and SPF mice during the light and dark phase quantified by flow cytometry. Data underlying this figure are supplied in S1 Data. (TIF)

**S5 Fig. Locomotor activity and total amount of cecal SCFAs.** (A-B) Locomotor activity in OligoMM12 and SPF mice ($n = 9$ per group): (A) Circadian variation in average breaks/minute per time point. (B) Average daily breaks/minute. (C) Estimation total amount of SCFAs and intermediate metabolites by multiplying measured concentration values by the cecal mass of the group. Number represented estimate mean value ± combined standard uncertainty from measurements used for calculations. Number of mice per group in all figures unless otherwise specified: GF = 13, OligoMM12 = 12, SPF = 10. $p$-values obtained by Tukey's honest significance test. Data underlying this figure are supplied in S1 Data. GF, germ-free; SCFA, short-chain fatty acid; SPF, specific-opportunistic-pathogen-free. (TIF)

**S6 Fig. Community composition of the OligoMM12 microbiota in cecum content during the light and dark phase quantified by 16S amplicon sequencing.** Data underlying this figure are supplied in S1 Data, and raw sequencing data are publicly available on the European Nucleotide Archive (ENA) under the Project ID PRJEB53981. (TIF)

**S7 Fig. Metabolic profile comparison of GF, OligoMM12, and SPF C57B6/J mice by UPLC/MS in liver.** Manually curated list of compounds obtained by targeted peak extraction from differentially expressed pathways in liver samples during the light phase (ZT 5) and dark phase (ZT 16). $p$-values obtained by Tukey's honest significance test after $\log_2$ transformation of area value. Number of mice per group: ZT5: GF = 4, OligoMM12 = 6, SPF = 7; ZT16: GF = 4, OligoMM12 = 6, SPF = 7. Data underlying this figure are supplied in S1 Data. GF, germ-free; SPF,

specific-opportunistic-pathogen-free; UPLC/MS, ultraperformance liquid chromatography coupled with mass spectrometry; ZT, Zeitgeber time.
(TIF)

**S8 Fig. Metabolic profile comparison of GF, OligoMM12, and SPF C57B6/J mice by UPLC/ MS in plasma.** Manually curated list of compounds obtained by targeted peak extraction from differentially expressed pathways in plasma samples during the light phase (ZT 5) and dark phase (ZT 16). *p*-values obtained by Tukey's honest significance test after log2 transformation of area value. Number of mice per group: ZT5: GF = 4, OligoMM12 = 7, SPF = 7; ZT16: GF = 5, OligoMM12 = 6, SPF = 6. Data underlying this figure are supplied in S1 Data. GF, germ-free; SPF, specific-opportunistic-pathogen-free; UPLC/MS, ultraperformance liquid chromatography coupled with mass spectrometry; ZT, Zeitgeber time.
(TIF)

**S1 Table. List of metabolites identified by targeted peak extraction in the UPLC/MS data.** Table indicates compound name, KEGG entry number, type of column was used for UPLC and if the peak ID matched the retention time and MS2 spectra identified with the chemical standard in liver and plasma samples. Data of all compounds in liver and plasma samples during the light phase (ZT 5) and dark phase (ZT 16) available in S1 Data. UPLC/MS, ultraperformance liquid chromatography coupled with mass spectrometry; ZT, Zeitgeber time.
(DOCX)

**S1 Data. Excel spreadsheet containing, in separate sheets, the underlying numerical data and statistical analysis for Figs 1C–1G, 2A–2I, 3A–3F, 4A–4F, S1A–S1D, S2A–S2K, S3A– S3E, S4, S5A–S5C, S6, S7, and S8.**
(XLSX)

# Acknowledgments

We would like to thank Thomas Fehr, Andre Galhano, and Susanne Freedrich for their support in the establishment of the gnotobiotic metabolic phenotype facility in the ETH Phenomic Center. Also, we thank Maria L. Balmer for her comments and suggestions for the manuscript.

# Author Contributions

**Conceptualization:** Daniel Hoces, Wolf-Dietrich Hardt, Christian Wolfrum, Emma Slack.

**Formal analysis:** Daniel Hoces, Jiayi Lan, Wenfei Sun, Tobias Geiser, Melanie L. Stäubli, Markus Arnoldini.

**Funding acquisition:** Andrew J. Macpherson, Shinichi Sunagawa, Wolf-Dietrich Hardt, Christian Wolfrum, Emma Slack.

**Investigation:** Daniel Hoces, Jiayi Lan, Wenfei Sun, Melanie L. Stäubli, Elisa Cappio Barazzone, Tenagne D. Challa, Manuel Klug, Alexandra Kellenberger, Sven Nowok, Erica Faccin.

**Methodology:** Daniel Hoces, Jiayi Lan, Wenfei Sun, Tobias Geiser, Melanie L. Stäubli, Markus Arnoldini, Shinichi Sunagawa, Renato Zenobi.

**Project administration:** Daniel Hoces, Emma Slack.

**Resources:** Bärbel Stecher, Shinichi Sunagawa, Renato Zenobi, Wolf-Dietrich Hardt, Christian Wolfrum, Emma Slack.

**Supervision:** Shinichi Sunagawa, Wolf-Dietrich Hardt, Christian Wolfrum, Emma Slack.

**Visualization:** Daniel Hoces, Jiayi Lan, Tobias Geiser, Melanie L. Stäubli, Emma Slack.

**Writing – original draft:** Daniel Hoces, Emma Slack.

**Writing – review & editing:** Daniel Hoces, Jiayi Lan, Wenfei Sun, Tobias Geiser, Melanie L. Stäubli, Elisa Cappio Barazzone, Markus Arnoldini, Tenagne D. Challa, Manuel Klug, Alexandra Kellenberger, Sven Nowok, Erica Faccin, Andrew J. Macpherson, Bärbel Stecher, Shinichi Sunagawa, Renato Zenobi, Wolf-Dietrich Hardt, Christian Wolfrum, Emma Slack.

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
