## [Editor Report · Decision Letter 0]

21 Dec 2021

Dear Dr Slack, 

Thank you for submitting your manuscript entitled "Metabolic reconstitution by a gnotobiotic microbiota varies over the circadian cycle" for consideration as a Research Article by PLOS Biology.

Your manuscript has now been evaluated by the PLOS Biology editorial staff, as well as by an academic editor with relevant expertise, and I am writing to let you know that we would like to send your submission out for external peer review.

Once your full submission is complete, your paper will undergo a series of checks in preparation for peer review. Once your manuscript has passed the checks it will be sent out for review. To provide the metadata for your submission, please Login to Editorial Manager (https://www.editorialmanager.com/pbiology) within two working days, i.e. by Dec 22 2021 11:59PM.

If your manuscript has been previously reviewed at another journal, PLOS Biology is willing to work with those reviews in order to avoid re-starting the process. Submission of the previous reviews is entirely optional and our ability to use them effectively will depend on the willingness of the previous journal to confirm the content of the reports and share the reviewer identities. Please note that we reserve the right to invite additional reviewers if we consider that additional/independent reviewers are needed, although we aim to avoid this as far as possible. In our experience, working with previous reviews does save time. 

If you would like to send previous reviewer reports to us, please email me at dummarino@plos.org to let me know, including the name of the previous journal and the manuscript ID the study was given, as well as attaching a point-by-point response to reviewers that details how you have or plan to address the reviewers' concerns. 

Given the disruptions resulting from the ongoing COVID-19 pandemic, please expect some delays in the editorial process. We apologise in advance for any inconvenience caused and will do our best to minimize impact as far as possible. In addition, please note that we will invite reviewers from January, as the likelihood of having well placed reviewers decline during the upcoming festive break is high. 

Kind regards,

Dario

Dario Ummarino, PhD

Senior Editor

PLOS Biology

---

## [Decision Letter · Decision Letter 1]

1 Mar 2022

Dear Dr Slack,

Thank you for submitting your manuscript "Metabolic reconstitution by a gnotobiotic microbiota varies over the circadian cycle" for consideration as a Research Article at PLOS Biology. Your manuscript has been evaluated by the PLOS Biology editors, an Academic Editor with relevant expertise, and by several independent reviewers.

As you will see in the reviews attached below, the reviewers appreciate the rigorous experimental set-up of the study, in particular the use of the metabolomic cage system; they also think that your findings would be valuable to the microbiology/metabolism research community. However, the reviewers also raise several points to improve the interpretation and discussion of the results. In particular, we would like to emphasize that you should carefully address the concerns around locomotion data raised by reviewers #1 and #2, as these data would be important to fully explain RER differences. Please also pay close attention to the additional analyses on the circadian microbiome dynamics in the OligoMM12 group, which were requested by reviewers #2 and #3. We also recommend that you provide evidence for sterility after continuous housing, but not necessarily at the end of the experiments, to address the comments by reviewer #2 and #3. Finally, in addressing the comments by reviewer #2, we recommend that you move most of the results related to cecum removal to the supplementary material and you shorten the discussion of these results, while on the other hand you elaborate more on the expected effects on bile acids.

In light of the reviews, we will not be able to accept the current version of the manuscript, but we would welcome re-submission of a much-revised version that takes into account the reviewers' comments. We cannot make any decision about publication until we have seen the revised manuscript and your response to the reviewers' comments. Your revised manuscript is also likely to be sent for further evaluation by the reviewers.

Given the potential time frame for performing the additional experiments requested by the reviewers, we expect to receive your revised manuscript within 6 months, although you can submit the revisions as soon as they are ready. 

**IMPORTANT - SUBMITTING YOUR REVISION**

*Re-submission Checklist*

*Published Peer Review*

*PLOS Data Policy*

*Blot and Gel Data Policy*

Sincerely,

Dario

Dario Ummarino, PhD

Senior Editor

PLOS Biology

dummarino@plos.org

REVIEWS:

Reviewer #1: Hoces et al. present a careful examination of how OligoMM12 mice differ from SPF and GF mice in terms of bodyweight (Fig. 1), energy balance (Fig. 2), RER and SCFAs (Fig. 3), and metabolomics of liver and plasma (Fig. 4). They use a clever setup to monitor calorimetric measurements under sterile conditions, which allows them to address some of the long-standing questions in the field of metabolic host-microbiota interactions.

Among the most interesting outcomes are the finding that OligoMM12 mice have larger fat depots than both SPF and GF mice (Fig. 1G), that GF mice successfully compensate for their lack of microbiota-derived calories by eating more (Fig. 2), and that GF, SPF, and OligoMM12 mice have approximately equivalent energy expenditure (Fig. 2B) and absolute energy extraction (Fig. 2K), but SPF mice extract a higher percentage of calories from food (Fig. 2L). This manuscript contributes to an improved understanding of the ways in which OligoMM12 mice do and don't mimic SPF mice.

Please find below several questions and suggestions that I hope will help improve this manuscript.

* Line 124: The authors present absolute lean mass in Fig. 1F, but it would be nice to see percent lean mass. For example, SPF mice seem to have the lowest body weight (Fig. 1D), but the highest lean body mass (Fig. 1F), suggesting that they have significantly higher percent lean mass. This would be an interesting finding to report. The same is true for percent fat mass, but I understand the authors' reservations about doing this given their observation that the measurement of fat mass is sensitive to cecal weight (Fig. S1E).

* Line 128: It is an interesting point that EchoMRI measurements are different pre- and post-cecum dissection. In the Methods, I saw that "for a set of mice," EchoMRI was performed on euthanized mice pre- and post-cecum removal. Are all the datapoints in Fig. S1E on dead mice, or is the y-axis dead mice and the x-axis live mice? I bring this up because the inconsistency in fat mass may also be related to comparing live v. dead mice. At least one publication reports EchoMRI inconsistencies when comparing live to dead animals (PMID: 21152249).

* Line 129: In Fig. S1, I suggest changing the word "variation" to "difference." Variation made me think standard deviation and variance, but the authors are simply referring to the difference between pre- and post-cecum measurements.

* Line 131: Fig. S1F is indeed baffling: how could there be higher fat mass when excluding the cecum? I appreciate that the authors included this puzzling result. They should discuss if this is incongruent with previously reported results.

* Line 134: The authors argue that directly weighing dissected fat mass is a more reliable way of estimating fat mass, given the EchoMRI weirdness they showed in Fig. S1E. However, dissection of small tissues weighing 50 to 500 mg (Fig. 1G) also seems imperfect. Very slight differences in the boundaries of tissue removal could have large effects on tissue weight. It was reassuring to read that dissection was performed while "blinded to the hygiene status of the mice," but, in my experience, the very large differences in cecum size prevent true blinding. One way to strengthen the authors' argument is to quantitatively compare their tissue weights to previously reported results (e.g. Bäckhed 2014, Suárez-Zamorano 2015), although such comparisons across studies and vivaria might be difficult.

* Line 150: Did the authors also try normalizing by lean body mass after subtracting the weight of the cecum? The cecum should be mostly (entirely?) lean mass, rather than fat mass, so this normalization approach would mimic the "classical normalization" of dividing by lean body mass but after adjusting for the confound of cecal weight.

* Line 165: It is interesting and surprising that feces from GF mice have fewer calories than feces from SPF mice. I like the authors' explanation for this observation, but they should discuss it in the context of previous papers that have reported bomb calorimetry done on feces from gnotobiotic mice.

* Line 203: Showing that changes in food intake don't explain differences in RER is an important control. The other behavior that could affect RER is locomotion, which unfortunately doesn't seem to have been measured in this manuscript. Physical activity is known to increase RER, so it would have been nice to investigate whether differences in physical activity could explain the differences in RER. Do the authors happen to have measurements of locomotion?

* Line 263: In Fig. 4C-D, the most consistent hit seemed to be butanoate metabolism. Are any of the molecules in Fig. S4 related to butanoate metabolism? It could be nice to have a sentence or two (and a graph or two) zooming in on this hit.

* Line 355: The RER data were used to emphasize that OligoMM12 mice resemble GF mice during the day and SPF mice at night, but the metabolomic data consistently showed that OligoMM12 and GF mice were always close together (both during the day and night). Therefore, I'm not sure that "In line with the RER data" is the appropriate transition here.

Praiseworthy highlights:

* Line 187: This is a nice result showing that SPF mice have microbes (not present in OligoMM12 mice) that improve energy extraction from food. This is consistent with OligoMM12 mice having a bigger cecum than SPF mice.

* Line 222: OligoMM12 mice producing much more hydrogen than the two other groups is a striking result.

* It is clear that the authors thought carefully about appropriate data normalization.

Very minor concerns:

* Line 246: Please define UPLC-MS the first time it's introduced.

* Typo in line 256: "OilgoMM12"

* Typo in line 454: "After samples all samples"

Reviewer #2: Hoces, et al. present a manuscript demonstrating the effects of GF and a reduced community, OligoMM12 on energy expenditure, RER, SCFA metabolism, and metabolomics changes. In particular the true contribution of the manuscript is the ability to use metabolic cages in an isolator system, thus allowing better metabolic characterization of GF and Oligo mice. This is truly remarkable and exciting as is seeing their data confirming and challenging hypotheses of the metabolic effects on GF and reduced community systems. Moreover, the authors recognize that cyclical changes in feeding and their dynamic effects on the luminal environment and host dynamics influences their metabolic measures, which again strengthens the paper. A few reservations exist though. Better circadian characterization of the microbiome of the Oligo and SPF would have helped inform bacterial-metabolic relationships the authors are using. There was a particular focus on cecal weight (taking up two figures as well as supplemental figures) which is not interesting since it did not affect their ultimate conclusions and distracted from other more important metabolic findings. Finally, the metabolomic results, particularly of bile acids, a far more important finding that is relevant to many groups, is essentially ignored. More detailed comments below.

MAJOR COMMENTS

1. Though it is clear that the microbiome has cyclical dynamics in SPF mice (based on previous papers the authors have cited) and the GF mice have no microbiome dynamics (since they don't have a microbiome), what is the dynamic changes in the microbiome of Oligo mice? Could the different results observed in day vs. night be the result of shifts in relative abundance of bacteria in their reduced community? Or do they occur despite a lack of cyclical oscillations?

2. The authors present so many different versions of their EE data that the message is getting lost in their attempts to show their rigor. Though normalization to body compartments (not total body weight) is common and recommended (PMID 20103710), there needs to be more clarity as to whether their results would be different if they had used lean mass vs lean mass without cecum. My understanding of the experimental conditions presented here is that using lean mass (as opposed to other body compartments or total body weight) makes most sense since GF and low abundance communities (e.g. OligoMM12) lead to poor nutrition states. The fact that cecal removal affects lean mass percentage and could potentially affect EE is interesting. But the authors show that their conclusions were not affected by cecum removal (Fig 2CDE). Hence, I don't really understand why this is such a big part of the paper. It seems like most of what is Fig 1 and 2 can go into supplemental figures (i.e., here are the EE results, cecal removal did not affect our conclusions) and they should focus on the main message here (that EE is not different between GF/OligoMM12/SPF). 

3. The authors should include mouse activity data over time.

4. Given the important of bacterial bile acid modifications on host physiology, what is the effect of OligoMM12 on bile acids? The authors hint that these are different and include it in a supplementary table, but this will require far more fleshing out. Since the organisms in the OligMM12 are known, their potential effect on bile acid pool should also be known. Are bile acid changes in line with what is expected? Or are certain bile acid changes occurring at certain times? The current manuscript as written has a missed opportunity here. 

5. Calorie consumed in daytime vs. nighttime by GF mice appears to be different than those reported by others (PMID 25891358). Since the authors used a metabolic cage system it is likely that their results are more accurate. However, the authors should discuss these results in context of previous findings.

6. The authors state that "although [GF, Oligo] can equally fill up hepatic glycogen storages at the end of the dark phase, GF and OligoMM12 deplete hepatic glycogen faster during the light phase." However, don't these results contradict their results in Fig 3A which shows that they are using non-glycogen (i.e., primarily fatty acid) sources of energy. How can the authors explain this discrepancy? 

7. Given how hard it is to bring a metabolic cage system into a germ-free environment, and this is a purported novelty in the paper, the authors should include in their supplementary figures quality control experiments that confirm that GF and OligoMM12 mice are indeed GF and OligoMM12 by the end of the experiment. 

8. Though the authors do a good job of placing their findings in context of previous studies, it's not immediately clear what the significance of Oligo treated mice being similar to GF mice in the dark but (slightly) similar to SPF mice during the day is (especially without knowing more about the dynamics of the Oligo microbiome).

MINOR COMMENTS

9. TSE systems allow continuous measurement of EE. Is there a reason why hourly EE is not presented (as it is for RER and H2)? Area under the curve can be used for continuous time data. There may be additional insights gained by hourly EE.

10. Much of what is stated in the introduction, particularly in the last paragraph, should be shifted to the discussion. In particular, the authors should point out the novelty of having a metabolic cage system in the isolator which distinguishes their work from the others cited here (from my review no other one mentioned has done this). 

11. As with the previous comment, description of cecal mass contribution to body composition is important but should likely be in results or methods rather than the introduction.

12. The figures were extremely hard to see. However, this is likely due to the PDF-merge function of the journal since the supplementary figures were exceptionally clear. This has been noted to the editor to improve their software.

13. The authors may want to use Canberra PCoA plots to look at distances between the metabolomics since this tool accounts for the relationship between metabolites (as opposed to plotting them as independent variables), which is increasingly being used instead of PCoA to determine differences between groups (https://github.com/mwang87/q2_metabolomics). 

Reviewer #3: In this manuscript, the authors established a novel isolator-housed metabolic cage system to measure metabolic activities in germ-free and gnotobiotic mice with minimal contamination risk. Using this system, the authors observed no significant difference in energy extraction and expenditure among germ-free and SPF mice and mice with a defined microbiota (OligoMM12). The work goes on to demonstrate that mice with different microbiota groups have different circadian rhythms of respiratory exchange ratio, suggesting that the microbiota has an impact on the type of respiratory substrates in metabolism at different phases of a day/night cycle. The authors further showed differences in fat and glycogen accumulation, hydrogen and short-chain fatty acid production and host metabolome profiles in different microbiota groups. 

Overall, the study is significant as it provides a new version of metabolic cage system for gnotobiotic research. It will be useful for understanding the roles of the microbiota in metabolic regulation. The authors identified multiple metabolic differences among the microbiota groups. How the metabolic reactions are connected and the molecular basis remain to be determined.

Specific comments:

1. Since the new system aims to minimize contamination for long-term examination, it would be important to provide results to show that germ-free mice remain sterile and OligoMM12 mice are not contaminated during and after experiment.

2. In most cases, the authors showed averages of readouts by the metabolic cage system, which were very clear and informative. The authors have also included raw time-course readouts in the supplemental table. It would be very helpful if the authors could plot the time-course values to show the rhythmicity and consistency of measurements across multiple day-night cycles.

3. The authors showed that OligoMM12 mice have more fat accumulation and different RER rhythmicity compared to germ-free and SPF mice. It would be very important if the authors could provide details of how they colonized the microbiota, such as how long the colonization was and what composition the microbiota was in their experiment. The gut microbiota is rhythmic and the rhythmicity may impact host rhythms. It would be helpful if the authors could examine the rhythm of OligoMM12 microbiota to gain some insight into the role of OligoMM12 microbiota in regulating host metabolic rhythms.

4. The authors estimated energy contribution by fecal microbiota using pre-determined parameters including bacterial mass, bacterial density in feces and energy stored in bacteria. Did OligoMM12 mice and SPF mice have similar bacterial density in their feces so that the same parameters could be used for the estimation?

5. The authors showed that multiple amino acid metabolic pathways were enriched in the comparisons between OligoMM12 and GF and between OligoMM12 and SPF. Have authors seen differential abundances of corresponding amino acids besides Glycine, Serine, Threonine and Glutamate, as shown in Sup Fig 4?

---

## [Editor Report · Decision Letter 2]

24 Jun 2022

Dear Dr. Slack,

Thank you for your patience while we considered your revised manuscript "Metabolic reconstitution by a gnotobiotic microbiota varies over the circadian cycle" for publication as a Research Article at PLOS Biology. This revised version of your manuscript has been evaluated by the PLOS Biology editors, and the Academic Editor.

Based on our Academic Editor's assessment of your revision, we are likely to accept this manuscript for publication, provided you satisfactorily address the following data and other policy-related requests.

1. DATA POLICY:

Regardless of the method selected, please ensure that you provide the individual numerical values that underlie the summary data displayed in the following figure panels as they are essential for readers to assess your analysis and to reproduce it: Figures 1CDEFG, 2ABCDEFGHI, 3ABCDEF, 4ABCDEF, and Supplementary figures S1AD, S2ABCDEFGHIJK, S3ABCDE, S4, S5ABC, S6, S7, S8. We appreciate that some of the raw data is already in the manuscript.

**Please also ensure that figure legends in your manuscript include information on where the underlying data can be found, and ensure your supplemental data file/s has a legend.**

2. We suggest a modification of the title: "Metabolic reconstitution by a gnotobiotic microbiota varies over the circadian cycle and resembles that of germ-free mice during the day". This is a suggestion, so please modify as you think it fits better.

We expect to receive your revised manuscript within two weeks.

*Published Peer Review History*

*Press*

Sincerely,

Paula

---

Senior Editor,

pjaureguionieva@plos.org,

PLOS Biology

---

## [Editor Report · Decision Letter 3]

6 Jul 2022

Dear Dr. Slack,

Thank you for the submission of your revised Research Article "Metabolic reconstitution of germ-free mice by a gnotobiotic microbiota varies over the circadian cycle" for publication in PLOS Biology. On behalf of my colleagues and the Academic Editor, Jotham Suez, I am pleased to say that we can in principle accept your manuscript for publication, provided you address any remaining formatting and reporting issues. These will be detailed in an email you should receive within 2-3 business days from our colleagues in the journal operations team; no action is required from you until then. Please note that we will not be able to formally accept your manuscript and schedule it for publication until you have completed any requested changes.

PRESS

Sincerely, 

Paula

---

Senior Editor

PLOS Biology
